# A webGIS Application to Assess Seawater Quality: A Case Study in a Coastal Area in the Northern Aegean Sea

**Dimitra Kitsiou \***, **Anastasia Patera** , **George Tsegas** and **Theodoros Nitis**

Laboratory of Environmental Quality & Geospatial Applications, Department of Marine Sciences, School of the Environment, University of the Aegean, 81100 Mytilene, Lesvos, Greece; apatera@marine.aegean.gr (A.P.); gtsegas@auth.gr (G.T.); theonitis@aegean.gr (T.N.)
\* Correspondence: dkit@aegean.gr; Tel.: +30-2251-036819

**Abstract:** The assessment of seawater quality in coastal areas is an important issue as it is related to the welfare of coastal ecosystems, a prerequisite for the provision of the related ecosystem services. During the last decades, marine eutrophication has become an important problem in coastal waters as a result of nutrient inputs increase. Consequently, there is need for appropriate methods and tools to assess the eutrophication status of seawater which should be user-friendly to coastal managers and support the adoption of effective plans for the protection and sustainable development of the coastal environment. In this framework, a user-friendly webGIS application has been developed and the Strait of Mytilene at the southeastern part of the Island of Lesvos in the NE Aegean Sea, Greece, was used as a case study. The methodology includes, as a first step, the evaluation of the accuracy of spatial interpolators widely applied in oceanographic studies for assessing the spatial distribution of relevant variables. The most appropriate interpolator revealed for each variable is subsequently applied for the production of the representative thematic layer. The second step involves the integration of the information from the optimal thematic layers representing the spatial distributions of the variables under study; as a result, a new thematic layer illustrating the eutrophication status of the study area is produced. The webGIS application is fully available via a web browser and provides a number of geoprocessing modules developed in Python which implement the user interface, the application of the interpolation analytical tasks, the statistical evaluation toolset and the integration of the optimal interpolated layers. Suggestions for further improvement of the proposed webGIS application are discussed.

**Keywords:** Geographical Information Systems; spatial analysis; inverse distant weighted method; interpolation; cross-validation; Aegean Sea

## 1. Introduction

Eutrophication is a natural process resulting from the enrichment of nutrients within a water body. Nutrient inputs to coastal waters are increasing in coastal areas, since almost 45% of the total population lives within the coastal zone of 150 km from the sea [1]. Human-caused enrichment is considered as the most common environmental problem that degrades seawater quality [2]. The main impacts are the disturbance of the ecosystem balance, the accelerated growth of primary production and the degradation of the value of ecosystem goods and services [3]. In economic terms, the effect of eutrophication in coastal areas has been estimated to have an annual cost of up to $1 billion in the European Union [4]. At European level, several Directives have been established in order to limit nutrient inputs to water bodies, such as the Urban Waste Water Directive (UWWD-91/271/EEC), the Water Framework Directive (WFD-2000/60/EC), the Marine Strategy Framework Directive (MSFD-2008/56/EC) [5] as well as the OSPAR and Helsinki Conventions. The need to follow the requirements of the Directives and regulations as well as the continuous increase of environmental pressure due to human activities related to the sea, has led the scientific

community to focus on the development of methodologies and tools for the assessment of seawater quality [6]. The latter include evaluation of different methods and application of the most accurate approaches [7–11], use of modern techniques [12–14], development of new and modification of already existing eutrophication scales [15] as well as introduction and use of relevant indicators [3,16,17]. Furthermore, the development of geo-informatics and especially of Geographical Information Systems (GIS) has led to the development of accurate geo-databases which support the application of appropriate spatial analysis methods and mapping techniques for illustrating the spatial distributions of variables relevant to seawater quality [9,18–21]. Further co-estimation of these variables to assess the eutrophication levels at different spatio-temporal scales [21–24] is considered very important because of the multi-parametric character of the eutrophication phenomenon [2] and the need for easily understood illustrations/maps of seawater quality, not only by the scientific community, but by decision-makers and administrators as well [25–27].

On the other hand, it is well-known that seawater sampling measurements represent fragmented spatial information; therefore, the study of the spatial distribution of the measured variables is not possible at the appropriate level of detail. In this framework, the application of interpolation methods has proved to be very effective, especially in areas where dense sampling sites networks is difficult to be implemented. An interpolation method is able to estimate the unknown data values where no measurements have been carried out. In literature, a wide range of interpolation methods are available [28] and a wide spectrum of efforts has been carried out so far to assess their performance, since it is a pre-requisite for the production of thematic maps of high accuracy [24,29,30]. At this point it should be noted that sampling density affects significantly the interpolation accuracy. A dense sampling network, though costly, leads to more accurate estimations than low sampling density which may underrepresent the spatial variation of the variable under study. Therefore, interpolation accuracy results should always be considered as relevant to the distribution and density of the sampling network.

Furthermore, the rapid growth of internet has contributed to the fast evolution of GIS technology regarding the architecture, technology and software used towards the development of webGIS applications [31,32]. webGIS technology has multiple advantages compared to traditional GIS which is very expensive and targets mainly specialized users. The web-based GIS may be used by a wide spectrum of users, even non-experts in GIS or without a deep knowledge of GIS techniques. The latter is an important advantage since people involved in the assessment of seawater quality are quite often non-experts in GIS though they recognize their capabilities and need in these studies. In addition, web-based GIS applications come usually with a user-friendly interface and can be accessed simultaneously by multiple users. Furthermore, the popularity of mobile technology devices resulted, apart from the widespread use of webGIS applications, to the evolution of public participation as well, known as "participatory GIS" [33,34]; the latter, empowers the communities by allowing access and sharing of geo-spatial information to different groups of users (i.e., authorities, stakeholders, the wider public), by mapping and analyzing spatial data sets and by promoting collaboration within the decision-making process [35].

The importance of the development of webGIS applications in the scientific areas related to the protection and monitoring of the marine environment as well as to coastal and marine management, has been documented in the literature where various approaches can be found. The latter could be grouped in those focusing mainly on the visualization, sharing and publishing of datasets and those where geoprocessing tools are developed and are available for further analysis and interpretation of datasets. Some indicative webGIS applications are given below; in [36] a Mid-Atlantic data portal is developed for supporting ocean planning, in [37] a publicly available atlas of marine uses and natural resources is available for the Adriatic Sea Region, in [38] a marine information sharing and publishing system is proposed for releasing marine information in real time and dynamically, in [39] interactive visualization of marine pollution monitoring is possible via a web-based GIS, in [40] a webGIS for environmental monitoring of coastal areas influenced by the oil spill

industry is developed, in [41] a web-based system for monitoring coastal harmful algal blooms is presented integrating in situ observations, satellite data and numerical models and in [42] a GIS-based integrated framework for monitoring and forecasting coastal seiches is available.

In this paper, a user-friendly webGIS application is proposed for assessing seawater quality. The application includes a number of tools/modules allowing the user to perform accurate mapping of the eutrophication levels in the study area by co-estimating the spatial information from relevant variables. Assessment of the performance of different interpolators of the Inverse Distance Weighted method is possible using cross-validation. The co-estimation of the spatial information of relevant variables is performed by overlay with the possibility to assign weights to each variable. In addition, a case study is performed in the coastal area of the Strait of Mytilene in the North Aegean Sea, Greece using data collected during a sampling survey. The advantages of the proposed webGIS application, its potential users as well as the possibility to improve its functionality are presented.

## 2. Materials and Methods

### 2.1. Study Area and Dataset

The study area is the Strait of Mytilene (Figure 1) located in the North Aegean Sea in the south-eastern part of the Island of Lesvos with an extent of approximately 129 km$^2$. The administrative capital city of the island is Mytilene with a permanent population of about 27,500 inhabitants while there are almost 40,000 inhabitants in the total watershed area according to the Hellenic Statistical Authority. Facilities of wastewater treatment plants and sewage operate in Mytilene since 2001 covering the 98% of the capital's needs by sewerage according to the Greek Ministry of Environment and Energy. The smaller villages located in the study area are not included in the facilities' services, remaining in the use of cesspools. In addition, there is a large number of natural areas and a significant total area of agricultural activities according to the information obtained from the Corine Land Cover.

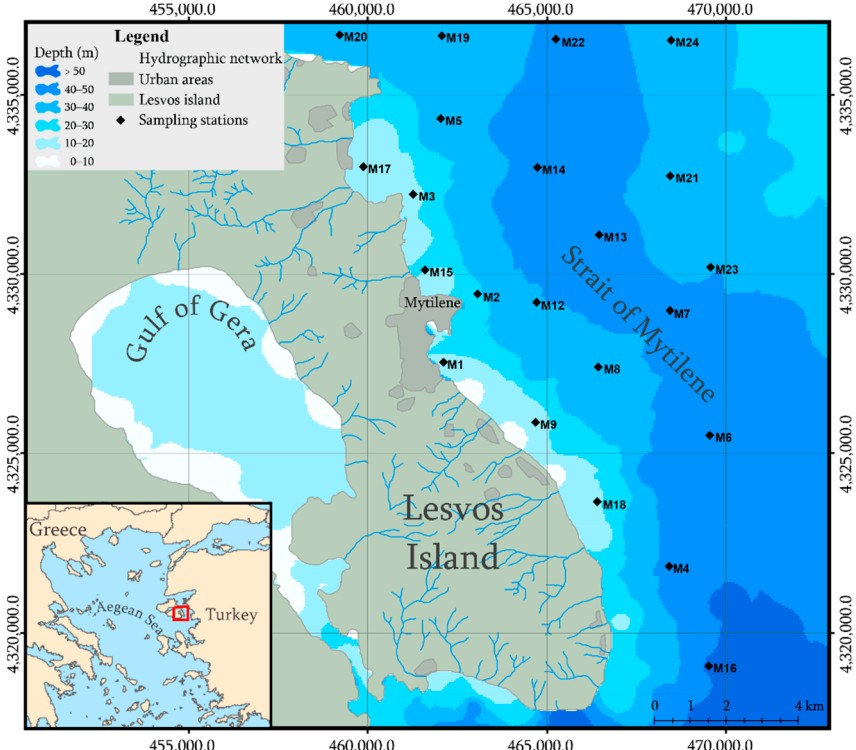

**Figure 1.** The Strait of Mytilene and the sampling sites location (Coordinate system: Universal Transverse Mercator/Zone 35).



The dataset used consists of data collected during field trips carried out in December 2007 in the framework of an Interreg III GR-CY program (2007–2009). Seawater samples were collected at 1 m depth from 22 sampling sites and the concentrations of chlorophyll *a* (chla, µg/L), dissolved nitrates (N-NO$_3$, µmol N/L) and ammonium (N-NH$_4$, µmol N/L) were measured. Dissolved inorganic phosphorous (P-PO4, µmol P/L) was measured as well; however, since the values indicated no significant variance being close to the detection limit of the method used, this variable was been included in the dataset. The sampling sites network was designed based on previous surveys in the area and the compilation of existing information regarding point-source pollution that is easily identified as it comes from a single place and nonpoint-source pollution that is not easily identified as it comes from many places all at once i.e., land runoff (Figure 1). This study area and the available data set were ideal for the development of the proposed webGIS application since various activities as well as several sources of pollution are met in the coast that can cause eutrophication episodes near the coastline.

### 2.2. Methodology

2.2.1. Spatial Interpolation Method

The Inverse Distance Weighted Interpolation method (IDW), initially described in [43], is a simple and fast method, applied very often in oceanographic data. The IDW method implements the assumption that a variable changes gradually into space so that nearby points have close values. Based on this principle, the nearest sampling points will have a higher contribution to the estimation of a simulated value than a distant sampling point, according to a weighting function. The formula of the IDW method is the following:

$$Z(x,y) = \frac{\sum\limits_{i=1}^{n} w(d_i)z_i}{\sum\limits_{i=1}^{n} w(d_i)} \tag{1}$$

where $Z(x,y)$ the simulated value of the variable at the point $(x,y)$, $w(d_i)$ the weighting function, $z_i$ the measured value at point $i$, $d_i$ the Euclidian distance of the point $i$ from the point $(x,y)$ and $n$ the number of the neighboring measured values considered. In this paper, $w(d) = 1/d^r$ where $r$ = 1, 2, 3, . . . .

The IDW method works well when the sampling points are distributed in the study area than when they are clustered. The number of the neighboring points $n$ as well as the degree of their influence (weighting function) can give the IDW interpolator more local character; i.e., a small number of neighbors $n$ combined with a large power value $r$.

In this paper, different interpolators of the IDW method were evaluated. The focus on this method was decided, since it is widely applied as a routine interpolation method in oceanographic studies due to its simplicity, rapid processing and the accurate results it provides; evaluations that compared IDW to other interpolation methods using oceanographic and marine datasets, proved the high performance of the IDW interpolators [44–47].

2.2.2. Evaluation Method

The cross-validation is a common and widely used technique for assessing the accuracy of interpolation methods [48–50], and has already been used to evaluate the performance of IDW interpolators [51,52]. The process involves the calculation of the residual error $e_i$ that is the deviation between the observed value $o_i$ of a variable at a specific location and the predicted value $p_i$ by the interpolation method. For this purpose, the first measurement from the dataset is removed and the remaining data are used to predict the value at that location. The process is repeated as many times as the available measurements and when it is completed all the measured values have been removed in turn and the relative predicted ones have been calculated.

The comparison of interpolators' performance has been usually considered in terms of several error metrics [29,53,54]. In this paper, five statistical metrics (Table 1) were used to assess the performance of IDW interpolators.

**Table 1.** Metrics to assess the performance of IDW interpolators.

| | | |
|---|---|---|
| Root mean square error (RMSE) | $\text{RMSE} = \sqrt{\frac{1}{n}\sum\limits_{i=1}^{n}\lvert e_i \rvert^2}$ | (2) |
| Normalized mean square error (NMSE) | $\text{NMSE} = \frac{1}{n}\sum\limits_{i=1}^{n}\frac{e_i^2}{\overline{p}\cdot\overline{m}}$ | (3) |
| Mean bias error (MBE) | $\text{MBE} = \dfrac{\sum\limits_{i=1}^{n} e_i}{n}$ | (4) |
| Mean absolute error (MAE) | $\text{MAE} = \dfrac{\sum\limits_{i=1}^{n}\lvert e_i \rvert}{n}$ | (5) |
| Index of agreement (IOA) | $\text{IOA} = 1 - \dfrac{\sum\limits_{i=1}^{n} e_i^2}{\sum\limits_{i=1}^{n}\left(\lvert p_i-\overline{m}\rvert+\lvert m_i-\overline{m}\rvert\right)^2}$ | (6) |

*n*: the number of the observed values, *m*: the observed values, *p*: the predicted values, $\overline{m}$: the mean of the observed values, $\overline{p}$: the mean of the predicted values, $e_i$: the deviation between the observed value and the predicted value, $e_i = p_i - o_i$.

The Root Mean Square Error (RMSE) is a statistical metric that is very often used for the assessment of the accuracy of predictions in combination with the MAE, MBE or other metrics [51,55,56]. RMSE is expected to be lower than MAE [57]. It has high sensitivity to outliers when the errors do not follow the normal distribution and seems to be more appropriate when the error distribution is expected to be Gaussian [57,58].

The Mean Bias Error (MBE) represents the average error when the signs of the errors are not removed and it is often used to indicate average model bias [59]. The typical error magnitude is often underestimated since the overestimated prediction at a point cancels the underestimated prediction at another. As a result, this metric is better when used in combination with other statistical metrics [60].

The Normalized Mean Square Error (NMSE) is a measure of the mean relative scatter and reflects the random errors [61]. The normalization of the MSE assures that the metric will not be biased when the model overestimates or underestimates the predictions. When the NMSE is higher than 1, the error distribution is rather log-normal than Gaussian [62].

The index of agreement [63] is described as the ratio of the mean square error and the potential error. It is a non-dimensional and bounded metric between 0 and 1; where 0 indicates no agreement and 1 the best agreement [64,65]. Its main drawback is that it has high sensitivity to larger than smaller deviations due to calculation as the squared values of the differences between the observed and predicted values [66].

In this paper, the accuracy assessment of five IDW interpolators was performed by co-estimation of the five above mentioned metrics (Table 1).

2.2.3. Seawater Quality

Seawater quality is directly related to the eutrophication status of a study area [67]; therefore, the assessment of the eutrophication levels in the area is considered of high importance. Furthermore, since marine eutrophication is a multi-parametric phenomenon, its assessment requests the co-estimation of more than one relevant variables [2]. In this paper, the information acquired of three variables was integrated for this purpose. As a first step, the best IDW interpolator resulting from the evaluation process was applied in order to convert the initial point dataset of each variable to a raster layer. Raster is a data model used in GIS which is represented by regularly-size rectangular or square shaped grid cells arranged in rows and columns. A raster layer is a set of raster data representing a particular geographic area. Pixels are the grid cells that make up rasters, are identical in size and represent the smallest unit of information in a raster. The pixel values of each raster were then classified (grouped) to four eutrophication levels, which characterize the

eutrophication status of the study area, according to the scales shown in Table 2; in these eutrophication scales, the boundary values between different eutrophication levels are defined. Furthermore, a code number (an integer) was assigned to the pixels of each class (pixels in a raster that represent the same condition) (Table 2). The next step was the overlay (a GIS operation that superimposes multiple data sets together) of the three rasters and the production of a final raster $R_f$ based on the following formula:

$$R_f = w_1 \cdot R_{\text{chla}} + w_2 \cdot R_{\text{N-NO}_3} + w_3 \cdot R_{\text{N-NH}_4} \tag{7}$$

where $R_{\text{chla}}$ the raster of the spatial distribution of the variable chla, $R_{\text{N-NO}_3}$ the raster of the spatial distribution of the variable N-NO$_3$, $R_{\text{N-NH}_4}$ the raster of the spatial distribution of the variable N-NH$_4$ and $w_1, w_2, w_3$ the weights assigned to each variable, respectively.

**Table 2.** Eutrophication scales for chla, N-NO$_3$ and N-NH$_4$ [15,25] and code numbers assigned to the eutrophication levels.

| Eutrophication Level | chla (µg/L) | N-NO$_3$ (µg-at N/L) | N-NH$_4$ (µg-at N/L) | Code Number |
|---|---|---|---|---|
| Oligotrophic | 0–0.10 | 0–0.62 | 0–0.55 | 1 |
| Lower mesotrophic | 0.10–0.60 | 0.62–0.65 | 0.55–1.05 | 2 |
| Upper mesotrophic | 0.60–2.21 | 0.65–1.19 | 1.05–2.20 | 3 |
| Eutrophic | 2.21< | 1.19< | 2.20< | 4 |

The only restriction is that the sum of the weights should be equal to 9; this number was chosen for three reasons: (i) the calculation of the boundaries of the eutrophication levels in Table 3 is facilitated, (ii) the assignment of equal weights to the variables is possible $w_1 = w_2 = w_3 = 3$ and (iii) the assignment of higher weight to one or two of the variables is also possible (i.e., $w_1 = 4, w_2 = 4, w_3 = 1$). The classification of the pixel values of the $R_f$ raster to the four eutrophication levels is based on a new scale developed for this purpose, as shown in Table 3. Since the sum of the weights should be equal to 9, it could be considered that 9 layers are overlayed with pixel values from 1 to 4 according to Table 2. Therefore, the minimum and the maximum values of the new scale are 9 and 36, corresponding to the oligotrophic and the eutrophic field, respectively. The threshold values of the eutrophication levels are calculated based on the principle that the level assigned to a pixel of the $R_f$ raster will be the one that characterizes the majority of the overlayed pixels of the representative layers. For example, as shown in Table 3, the value 22 of the new scale is the result of the overlay of 5 pixels characterized as lower mesotrophic (value 2) and 4 pixels characterized as upper mesotrophic (value 3); therefore, the value 22 corresponds to the lower mesotrophic field. Consequently, as shown in Table 3, pixel values of the $R_f$ raster from 9 to 13 are representative of the oligotrophic, 14 to 22 of the lower mesotrophic, 23 to 31 of the upper mesotrophic and 32 to 36 of the eutrophic field.

It should be noted that the final eutrophication map is a vector file, since after the overlay of the three rasters, the pixels with the same value are merged to a single polygon. An area corresponding to a particular eutrophication level may include more than one polygon, since the values of each polygon are classified according to the eutrophication scale (Table 3).

**Table 3.** Eutrophication scale for the classification of the pixel values of the $R_f$ raster that is the result of the integration of the three variables when weights are assigned to them.

| Eutrophication level | Overlayed Layers | | | | | | | | | $R_f$ raster (pixel values) |
|---|---|---|---|---|---|---|---|---|---|---|
| | 1st | 2nd | 3rd | 4th | 5th | 6th | 7th | 8th | 9th | |
| | Possible pixel values | | | | | | | | | |
| Oligotrophic | 1 | 1 | 1 | 1 | 1 | 1 | 1 | 1 | 1 | 9 |
| | 1 | 1 | 1 | 1 | 1 | 2 | 2 | 2 | 2 | 13 |
| Lower mesotrophic | 1 | 1 | 1 | 1 | 2 | 2 | 2 | 2 | 2 | 14 |
| | 2 | 2 | 2 | 2 | 2 | 2 | 2 | 2 | 2 | 18 |
| | 2 | 2 | 2 | 2 | 2 | 3 | 3 | 3 | 3 | 22 |
| Upper mesotrophic | 2 | 2 | 2 | 2 | 3 | 3 | 3 | 3 | 3 | 23 |
| | 3 | 3 | 3 | 3 | 3 | 3 | 3 | 3 | 3 | 27 |
| | 3 | 3 | 3 | 3 | 3 | 4 | 4 | 4 | 4 | 31 |
| Eutrophic | 3 | 3 | 3 | 3 | 4 | 4 | 4 | 4 | 4 | 32 |
| | 4 | 4 | 4 | 4 | 4 | 4 | 4 | 4 | 4 | 36 |

### 2.2.4. The webGIS Application

Typically, the interface of a webGIS application is a browser window on a personal computer where the users can interact with the application. The application presented in this paper was developed to provide a user-friendly interface with simple architecture and host a number of geoprocessing tools (Figure 2). For this purpose, the ArcGIS 10.2.2 ESRI and the ArcGIS for Server 10.2.2 were used to develop and store the GIS services. The Web Server of the ArcGIS Online was used for the distribution of the information offered by the webGIS application. Further technical details regarding specific webGIS features and components are out of the scope of this paper and can be found in literature [68].

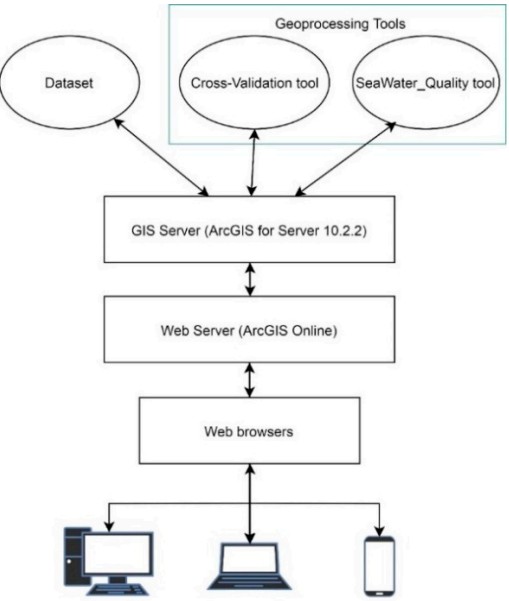

**Figure 2.** The architecture of the webGIS application.

Users have access to the application through internet connection using a web browser. The webGIS application allows the user (a) to visualize the content of the dataset where the location of the sampling sites and the measurements of the three variables are stored, (b) to assess the accuracy of the IDW interpolators to be applied, (c) to choose the best IDW interpolator, (d) to integrate the three variables in order to assess the eutrophication levels in the study area and (e) to produce a final thematic map with the marine eutrophication levels.

The interface of the webGIS application includes three kind of tools: (a) navigation tools that facilitate the exploration of the area, (b) auxiliary tools that facilitate the visualization or interpretation of the datasets and (c) the geoprocessing tools: (c1) Cross-Validation and (c2) SeaWater_Quality, which were developed in Python.

The tools for map navigation, including the zoom in (+)/zoom out (−) widgets, are available at the left side of the browser window (Figure 3). Manipulation of the dataset is possible using the auxiliary tools; 'Layer List': the list of available layers as well as the attribute table of each layer can be visualized and 'Legend': the legend of each layer can be visualized.

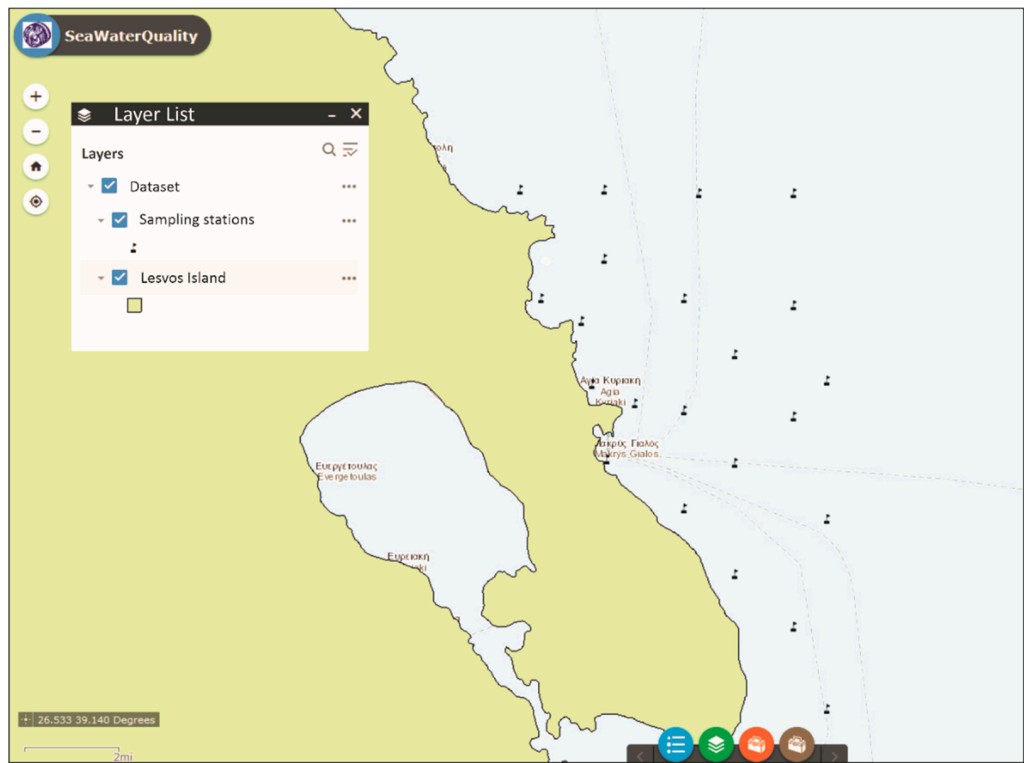

**Figure 3.** The user interface of the webGIS application (web link: https://wg-web1.aegean.gr/SeaWaterQuality/).

The Cross-Validation geoprocessing tool allows the evaluation and ranking of five IDW interpolators according to their accuracy. The tool after calculating the value of each metric of Table 1 ranks the interpolators according to their performance for each metric (1: for the best interpolator, 5: for the worst interpolator). Then, the ranking numbers of each interpolator are summed up and the lower the score the better the overall performance of the interpolator. As a result, the tool provides a table where the rank of the tested IDW interpolators and the calculated metrics for each one of them are stored. The Cross-Validation geoprocessing tool is recommended to be applied as many times is necessary in order to assess the better IDW interpolator.

The SeaWater_Quality geoprocessing tool integrates the information of the studied variables and produces a final map where the eutrophication levels are clearly delineated. As a first step, the best IDW interpolator, derived from the application of the Cross-Validation geoprocessing tool, is applied for each variable. The n and r values (Equation (1)) are the input for this tool and as a result, three rasters (one for each variable) with a spatial resolution of 100 × 100 m are created. The spatial resolution of 100 × 100 m is considered appropriate based on the extent of the study area, the minimum distance of the sampling sites as well as the spatial behavior of the parameters under study. The overlay process of these rasters follows, as described in Section 2.2.3. It is important to note here that there is a validation check regarding the correct assignment of the weights to the

variables; if the weights do not meet the pre-requisite set (the sum of the weights be equal to 9), a black field is mapped on the study area.

## 3. Results and Discussion

The Cross-Validation geoprocessing tool was applied once for each variable. In the present study, the same pairs of *n*–*r* for the IDW interpolators were tested for all variables. The calculated statistical metrics for the evaluated IDW interpolators and the ranking order of each interpolator are presented in Table 4 for chla, in Table 5 for N-NO$_3$ and in Table 6 for N-NH$_4$. The detection of the best interpolator was based on the sum of the ranking order numbers; the best interpolator is the one with the lowest sum. As a result, the best IDW interpolator for chla was that with *n* = 4 and *r* = 2, for N-NO$_3$ with *n* = 4 and *r* = 1, and for N-NH$_4$ with *n* = 3 and *r* = 1. Regarding chla, higher *r* corresponded to better performances, while for N-NH$_4$ better performances were achieved using both low *n* and r. However, regarding N-NO$_3$ similar conclusions are not evident.

**Table 4.** The calculated statistical metrics for the evaluated IDW interpolators for **chla**, the ranking order of each interpolator and the sum of the ranking order numbers.

| n | r | RMSE | RO | NMSE | RO | MBE | RO | MAE | RO | IOA | RO | Sum of RO Numbers |
|---|-----|--------|----|--------|----|--------|----|--------|----|--------|----|------|
| 4 | 2 | 0.1219 | 1 | 0.0485 | 1 | 0.0194 | 4 | 0.0996 | 1 | 0.7270 | 2 | 9 |
| 3 | 2 | 0.1226 | 2 | 0.0491 | 2 | 0.0186 | 3 | 0.1010 | 2 | 0.7358 | 1 | 10 |
| 3 | 1.5 | 0.1246 | 3 | 0.0507 | 3 | 0.0184 | 2 | 0.1025 | 4 | 0.7251 | 3 | 15 |
| 3 | 1 | 0.1271 | 5 | 0.0528 | 5 | 0.0183 | 1 | 0.1043 | 5 | 0.7118 | 4 | 20 |
| 4 | 1 | 0.1255 | 4 | 0.0513 | 4 | 0.0196 | 5 | 0.1021 | 3 | 0.7019 | 5 | 21 |

RO: Ranking Order, RM: Root Mean Square Error, NMSE: Normalized Mean Square Error, MBE: Mean Bias Error, IOA: Index of Agreement.

**Table 5.** The calculated statistical metrics for the evaluated IDW interpolators for **N-NO₃**, the ranking order of each interpolator and the sum of the ranking order numbers.

| n | r | RMSE | RO | NMSE | RO | MBE | RO | MAE | RO | IOA | RO | Sum of RO Numbers |
|---|-----|--------|----|--------|----|---------|----|--------|----|--------|----|------|
| 4 | 1 | 0.1051 | 1 | 0.4569 | 1 | −0.0071 | 1 | 0.0711 | 1 | 0.4238 | 3 | 7 |
| 3 | 1 | 0.1110 | 2 | 0.5022 | 3 | −0.0049 | 2 | 0.0804 | 3 | 0.4404 | 1 | 11 |
| 4 | 2 | 0.1120 | 3 | 0.5014 | 2 | −0.0018 | 4 | 0.0780 | 2 | 0.4195 | 4 | 15 |
| 3 | 1.5 | 0.1147 | 4 | 0.5284 | 4 | −0.0025 | 3 | 0.0835 | 4 | 0.4288 | 2 | 17 |
| 3 | 2 | 0.1187 | 5 | 0.5573 | 5 | −0.0002 | 5 | 0.0869 | 5 | 0.4166 | 5 | 25 |

RO: Ranking Order, RM: Root Mean Square Error, NMSE: Normalized Mean Square Error, MBE: Mean Bias Error, IOA: Index of Agreement.

**Table 6.** The calculated statistical metrics for the evaluated IDW interpolators for **N-NH₄**, the ranking order of each interpolator and the sum of the ranking order numbers.

| n | r | RMSE | RO | NMSE | RO | MBE | RO | MAE | RO | IOA | RO | Sum of RO Numbers |
|---|-----|--------|----|--------|----|--------|----|--------|----|--------|----|------|
| 3 | 1 | 0.3226 | 1 | 0.5103 | 1 | 0.0356 | 5 | 0.2352 | 1 | 0.6261 | 1 | 9 |
| 3 | 1.5 | 0.3276 | 2 | 0.5299 | 2 | 0.0323 | 4 | 0.2358 | 2 | 0.6215 | 2 | 12 |
| 3 | 2 | 0.3333 | 4 | 0.5520 | 4 | 0.0296 | 3 | 0.2369 | 3 | 0.6162 | 3 | 17 |
| 4 | 1 | 0.3282 | 3 | 0.5423 | 3 | 0.0234 | 2 | 0.2503 | 5 | 0.5850 | 5 | 18 |
| 4 | 2 | 0.3362 | 5 | 0.5728 | 5 | 0.0204 | 1 | 0.2489 | 4 | 0.5858 | 4 | 19 |

RO: Ranking Order, RM: Root Mean Square Error, NMSE: Normalized Mean Square Error, MBE: Mean Bias Error, IOA: Index of Agreement.

Furthermore, RMSE, NMSE and MAE proved successful for indicating the best interpolator in all three cases, while MBE was successful only in the case of N-NO$_3$ and IOA only in the case of N-NH$_4$.

After the detection of the best interpolator the spatial distribution of each variable was produced. The final eutrophication map is the result of the application of the Sea-Water_Quality geoprocessing tool, where the spatial distributions of the three variables

are integrated by assigning weights to each one of them. In this paper, two case studies are presented. In the first one, all variables were considered of equal importance ($w_1 = w_2 = w_3 = 3$), while in the second one, chla was considered of higher importance compared to N-NO$_3$ and N-NH$_4$ ($w_1 = 5, w_2 = 2, w_3 = 2$), since it is referred as the most representative indicator of marine eutrophication and has been widely used to assess seawater quality [69]. The final map can be visualized in the browser window.

### 3.1. Case Study I: Equal Weights of Importance

In this case study, the three variables were assigned the same weights ($w_1 = w_2 = w_3 = 3$); therefore, the representative layers were overlayed according to the formula:

$$R_f = 3 \cdot R_{\text{chla}} + 3 \cdot R_{\text{N-NO}_3} + 3 \cdot R_{\text{N-NH}_4} \tag{8}$$

The result is illustrated in Figure 4 where lower mesotrophic field is observed mainly at the north-western part of the Strait of Mytilene including the sea area near the town of Mytilene and the south-eastern part. The rest of the study area is characterized as oligotrophic.

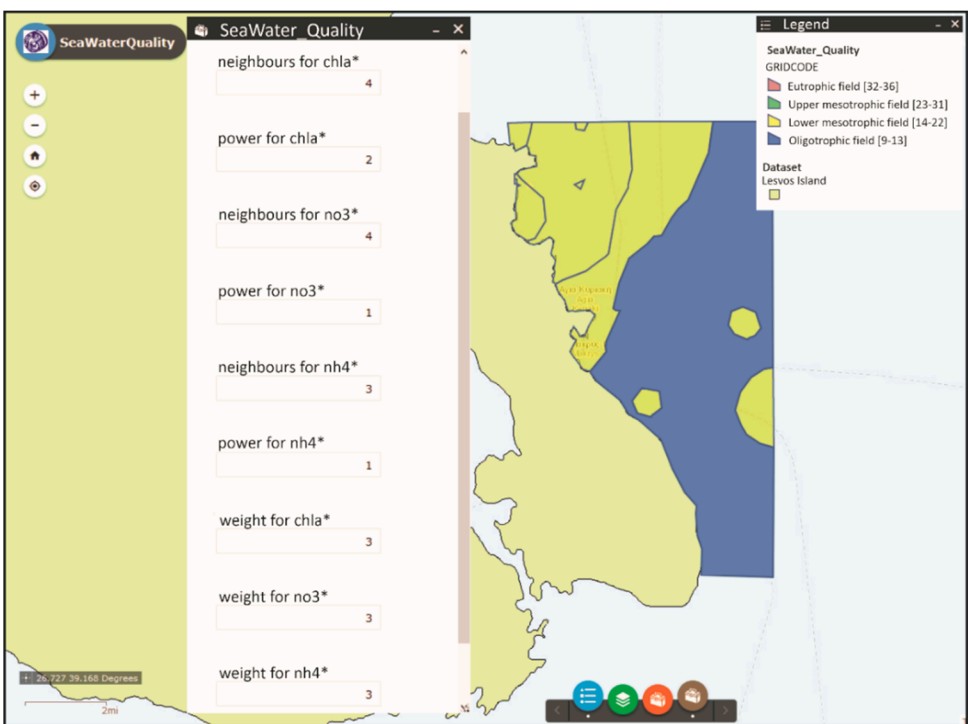

**Figure 4.** Spatial distribution of the eutrophication levels in the Strait of Mytilene when equal weights were assigned to all variables (The * indicates that these fields must be filled in by the user).

### 3.2. Case Study II: Higher Weight is Assigned to Chlorophyll a

In this case study, chla was assigned higher weight compared to the two other variables which were considered of equal importance ($w_1 = 5, w_2 = 2, w_3 = 2$). The formula used for the production of the final map is the following:

$$R_f = 5 \cdot R_{\text{chla}} + 2 \cdot R_{\text{N-NO}_3} + 2 \cdot R_{\text{N-NH}_4} \tag{9}$$

The eutrophication thematic map produced (Figure 5), indicates that all the study area is characterized as lower mesotrophic with only a very small area in the north which is upper mesotrophic.

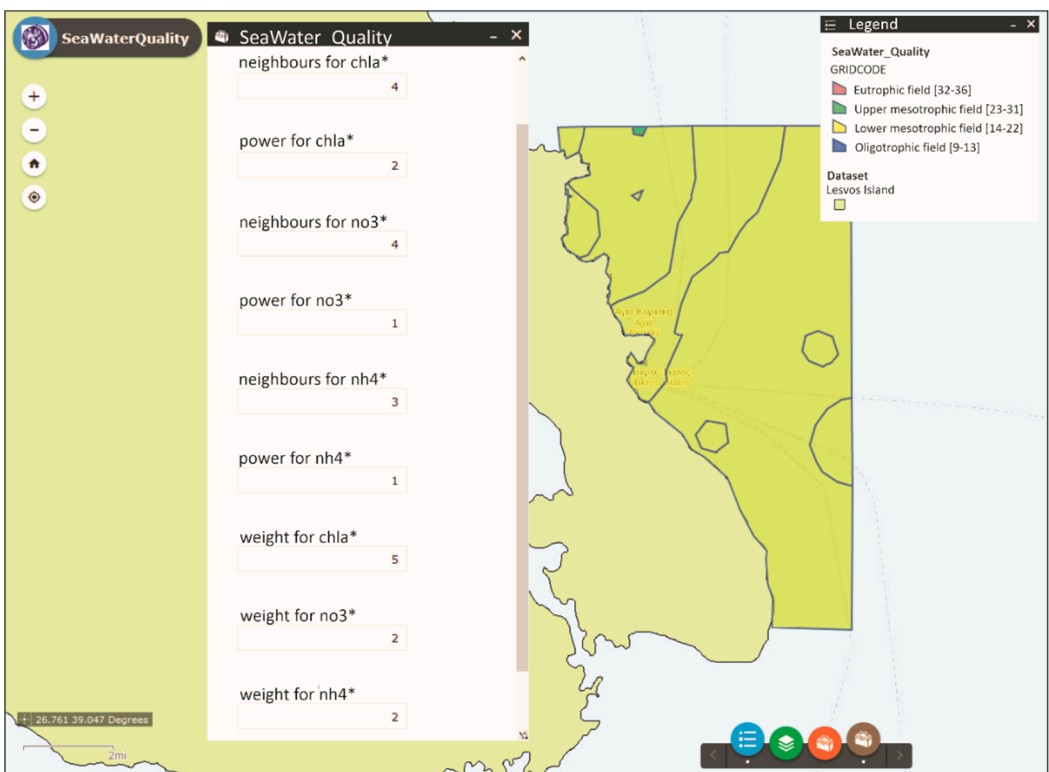

**Figure 5.** Spatial distribution of the eutrophication levels in the Strait of Mytilene when higher weight was assigned to chla. (The * indicates that these fields must be filled in by the user).

## 4. Conclusions

The assessment of seawater quality is of major importance in coastal areas, where various activities are met, and a prerequisite for decision-making and coastal zone management. In this context, thematic maps illustrating the spatial distribution of marine eutrophication where the eutrophication levels are clearly delineated, are needed. For this purpose, application of interpolation methods as well as integration of the spatial information acquired of a number of relevant variables are applied. The accuracy of the latter processes is reflected to the precise delineation of the eutrophication levels. Therefore, validation of the potential interpolators and selection of the best one for each studied variable is important. In addition, availability of user-friendly tools which would allow the effective implementation of the above mentioned procedures in a menu-driven environment by non-experts in GIS brings considerable added value. Further implementation of such tools in a webGIS environment enhances their accessibility by a variety of users. In this context, the webGIS application presented in this paper has a number of advantages: (a) Access is possible using only a simple web browser: Given that nowadays internet facilities are available to the wider public at a constantly increasing number of locations and types of devices, access to the application by users from various places is possible at minimal cost, without any restrictions concerning complex hardware infrastructures. (b) Awareness of any sophisticated GIS software is not needed by users: This is another important characteristic, since a high percentage of people involved in decision-making or simply interested in seawater quality are not computer experts and have limited GIS knowledge. (c) User-friendly system: The system could be characterized as user-friendly, since all operations are implemented via comprehensible interfaces allowing use by people not deeply familiarized with GIS. (d) Wide range of clients supported: A high number of users can use the application simultaneously. (e) The operation of the application is completely controlled: The system administrator is the only responsible for the proper operation of the webGIS application. (f) The application is possible to be applied multiple times with different options, ignoring one or more parameters if requested. This ability

gives also the opportunity to study the spatial distribution of each variable individually (by assigning zero weight to the other variables). (g) The performance of the IDW interpolators is assessed by combination of the results of five metrics, since the use of several statistical metrics, instead of only one or two, is highly recommended for the detection of the best interpolator. Additional metrics could be added if it is considered necessary. In addition, the current application proposes a methodology to detect the ranking order of interpolators according to each metric by synthesis of their individual rankings; therefore, all metrics are taken into account. (h) A wide range of possible IDW interpolators can be evaluated–various pairs of $n$ and $r$. In this paper, five IDW interpolators were selected to be assessed based on the number and the spatial distribution of the sampling sites. (i) There is simplicity and clarity in the illustration of the results which can be easily communicated to scientists and decision-makers making easier the exchange of ideas and submission of proposals regarding seawater quality. (j) The application is open to further development and extension of its functionalities, based on raised demands. The dataset can be updated and additional variables could be incorporated. Furthermore, the application could be adapted and become functional to other study areas.

Finally, this webGIS application could be incorporated to an already available web Decision Support System (DSS), where assessment of seawater quality is a prerequisite for management interventions, and represent a module focusing on exploratory data analysis. In this way, datasets, analysis results, maps, and services could be shared dynamically offering continuous communication among different groups of users. In general, this type of applications are very popular means of putting state-of-the-art visualization technology, spatial analysis and mapping techniques into the hands of scientists, local authorities, decision-makers, and stakeholders engaged in marine planning and interested in seawater quality and the protection of the marine environment.

**Author Contributions:** Conceptualization, D.K. and A.P.; methodology, D.K. and A.P.; web application, A.P. and G.T.; validation, A.P., G.T. and T.N.; writing—original draft preparation, A.P.; writing—review and editing, D.K. and T.N.; supervision, D.K. All authors have read and agreed to the published version of the manuscript.

**Funding:** This research received no external funding.

**Institutional Review Board Statement:** Not applicable.

**Informed Consent Statement:** Not applicable.

**Data Availability Statement:** Data sharing not applicable.

**Conflicts of Interest:** The authors declare no conflict of interest.

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
