# Peer review of "A webGIS Application to Assess Seawater Quality: A Case Study in a Coastal Area in the Northern Aegean Sea"

_jmse, doi:10.3390/jmse9010033_

Round 1

Reviewer 1 Report

The paper addresses the seawater quality monitoring via an application connected to internet. A main difficulty concerns the choice of the interpolation method between nodes where water quality measurements were available.

The introduction presents the web GIS challenge. The 2nd chapter gives the interpolation methods and the architecture of the application. The chapter 3 shows the study of a part of the Lesbos island coastal sea.

The paper is very well written and interesting to follow without heavy and complex concepts. Here follows minor suggestions and comments to improve the text.

“point and non-point” repeated lines 121 and 123 is unclear particularly ‘non-point’.

F(x,y) may be Z(x,y) in equation (1), it is not mandatory

Usage of both IDW interpolator and IDW interpolators (check lines 146, 150)

The lines 191-194 are unclear: raster; layer; pixel; levels; scale; code number; class; overlay. A definition of these terms can be given for clarity.

It is understood that 4*9 is the value 36 at line 204

First column format of Table 2 could be the same in Table 1.

Explanation of the italic characters lines 235-236 is necessary.

Characters in figure 3 are very small, even on the colour version (idem figures 4and 5)

“for each variables” is associated to Three variables, line 260, and Three times, line270. It may surprise the reader.

May be use Bold policy for the variables to distinguish Tables 4, 5 and 6

Weight suggested at line 299 is only given at line 319.

Polygon is used at line 304 for the first time could be used before.

The ‘non-expert’ concept appears in the conclusion at line 338. This concept may be considered as a central question and consequently be introduced carefully.

Check reference n°4

Author Response

Dear Reviewer 1,

thank you very much for your comments and suggestions. Please find below our response to your comments.

(1) "point and non-point” repeated lines 121 and 123 is unclear particularly ‘non-point’.

TEXT HAS BEEN ADDED IN LINES 129-134 OF THE NEW VERSION OF THE MANUSCRIPT IN ORDER TO CLARIFY POINT AND NON-POINT SOURCE  POLLUTION.

(2) F(x,y) may be Z(x,y) in equation (1), it is not mandatory.

IT HAS BEEN REPLACED.

(3) Usage of both IDW interpolator and IDW interpolators (check lines 146, 150)

'IDW interpolator' has been changed to 'IDW interpolators' IN LINE 160 OF THE NEW VERSION OF THE MANUSCRIPT.

(4) The lines 191-194 are unclear: raster; layer; pixel; levels; scale; code number; class; overlay. A definition of these terms can be given for clarity.

TEXT HAS BEEN ADDED IN LINES 206-214 OF THE NEW VERSION OF THE MANUSCRIPT IN ORDER TO DEFINE AND CLARIFY THESE TERMS.

(5) First column format of Table 2 could be the same in Table 1.

THE FORMAT OF TABLE 1 HAS BEEN CHANGED TO THAT OF TABLE 2.

(6) Explanation of the italic characters lines 235-236 is necessary.

THE ITALIC CHARACTERS HAVE BEEN REPLACED TO NORMAL.

(7) Characters in figure 3 are very small, even on the colour version (idem figures 4and 5).

FIGURES 3,4,5 HAVE BEEN IMPROVED.

(8) “for each variables” is associated to Three variables, line 260, and Three times, line270. It may surprise the reader.

IN LINE 291 OF THE NEW VERSION OF THE MANUSCRIPT THE 'three times' HAS BEEN DELETED.

(9) May be use Bold policy for the variables to distinguish Tables 4, 5 and 6.

BOLD POLICY HAS BEEN USED IN TABLES 4,5,6 FOR THE VARIABLES.

(10) Weight suggested at line 299 is only given at line 319.

IN LINES 319 AND 320 OF THE NEW VERSION OF THE MANUSCRIPT THE WEIGHTS ASSIGNED IN THE TWO CASE STUDIES HAVE BEEN ADDED IN THE TEXT.

(11) Polygon is used at line 304 for the first time could be used before.

THIS PARAGRAPH (LINES 322-326 IN THE NEW VERSION OF THE MANUSCRIPT) HAS BEEN MOVED TO LINES 236-239.

(12) The ‘non-expert’ concept appears in the conclusion at line 338. This concept may be considered as a central question and consequently be introduced carefully.

THE 'NON-EXPERT' CONCEPT HAS BEEN CLARIFIED IN THE INTRODUCTION (LINES 76-78).

(13) Check reference n°4

THIS REFERENCE HAS BEEN CORRECTED.

Reviewer 2 Report

The seawater quality of a coastal sector in the Northern Aegean was evaluated through a user-friendly webGIS application. In particular, based on two steps of control of variables accuracy and integration of information of spatial distributions of variables, the eutrophication status of the study area was obtained.

The manuscript is interesting and well structured. The figures and tables are clear. Metrics and math formulas seem to work well.

However, the input values and the related model are based only on 22 samples (see P3L115-120), that is 1 sample/2.5 km. A question is if more samples in the same area could obtain different results and assessments. This is an aspect to clarify in the Introduction section.

Anyway, a few changes are required to help readers understanding some aspects.

TEXT

P1L2: "sea water" or "seawater"? Check also in the whole text.

P1L35: percent CHANGE to %.

P3L107-108: "Hellenic Statistical Authority (2011)" not in the References. Add or delete it.

P3L113: "Corine Land Cover (2000)" not in the References. Add or delete it.

P7L112: "33 to 36" while in Table 2 there is 32 | 36, correct it.

P13L347-349: "User friendly system: ...  with GIS." Correct to "User-friendly". I understood what you mean, but the sentence is not clear. Rephrase it.

FIGURES

Figure 1: add the geographic coordinates (labeling) along the framework and indicate the coordinate system in the caption. To the left of the graphic scale, change "Km" to "km" (the lowercase letter k). No numbers for the samples?

Author Response

Dear Reviewer 2,

thank you very much for your comments and suggestions. Please find below our response to your comments.

(1) The input values and the related model are based only on 22 samples (see P3L115-120), that is 1 sample/2.5 km. A question is if more samples in the same area could obtain different results and assessments. This is an aspect to clarify in the Introduction section.

IN LINES 66-71 TEXT HAS BEEN ADDED IN ORDER TO CLARIFY THIS ISSUE.

(2) P1L2: "sea water" or "seawater"? Check also in the whole text.

"sea water" HAS BEEN CHANGED TO "seawater" THROUGHOUT THE WHOLE TEXT.

(3) P1L35: percent CHANGE to %.

IT HAS BEEN CHANGED.

(4) P3L107-108: "Hellenic Statistical Authority (2011)" not in the References. Add or delete it.

THE (2011) HAS BEEN DELETED AND THE WEBSITE OF THE "Hellenic Statistical Authority" HAS BEEN ADDED. LINE 115 IN THE NEW VERSION.

(5) P3L113: "Corine Land Cover (2000)" not in the References. Add or delete it.

THE (2000) HAS BEEN DELETED AND THE WEBSITE OF THE "Corine Land Cover" HAS BEEN ADDED. LINE 121 IN THE NEW VERSION.

(6) P7L112: "33 to 36" while in Table 2 there is 32 | 36, correct it.

IT HAS BEEN CORRECTED. LINE 232 IN THE NEW VERSION.

(7) P13L347-349: "User friendly system: ...  with GIS." Correct to "User-friendly". I understood what you mean, but the sentence is not clear. Rephrase it.

THE SENTENCE HAS BEEN RE-PHRASED. LINES 366-368 IN THE NEW VERSION.

(8) Figure 1: add the geographic coordinates (labeling) along the framework and indicate the coordinate system in the caption. To the left of the graphic scale, change "Km" to "km" (the lowercase letter k). No numbers for the samples?

THE GEOGRAPHIC COORDINATES ALONG THE FRAMEWORK HAVE BEEN ADDED. "Km" HAS BEEN CHANGED TO "km". THE NUMBERS OF THE SAMPLES (NAMES OF THE SAMPLING STATIONS) HAVE BEEN ADDED. THE COORDINATE SYSTEM HAS BEEN INDICATED IN THE CAPTION.